# Association of Four Dietary Patterns and Stair Climbing with Major Adverse Cardiovascular Events: A Large Population-Based Prospective Cohort Study

**DOI:** 10.3390/nu16213576

**Published:** 2024-10-22

**Authors:** Kexin Li, Yanqiu Huang, Liao Wang, Yong Yuan, Xu Jiang, Yang Yang, Qingyun Huang, Hui Wang

**Affiliations:** 1State Key Laboratory of Systems Medicine for Cancer, Center for Single-Cell Omics, School of Public Health, Shanghai Jiao Tong University School of Medicine, Shanghai 200025, Chinahuangyanqiu@sjtu.edu.cn (Y.H.); 2Shanghai Frontiers Science Center of Degeneration and Regeneration in Skeletal System, Shanghai Key Laboratory of Orthopaedic Implants, Department of Orthopaedic Surgery, Shanghai Jiao Tong University of Medicine affiliated Ninth People’s Hospital, Shanghai 200023, China; wangliaotaizhou@sjtu.edu.cn (L.W.); jiangxusun@foxmail.com (X.J.); 3Putuo Hospital, Shanghai University of Traditional Chinese Medicine, Shanghai 200333, China; 4Bloomberg School of Public Health, Johns Hopkins University, 615 N. Wolfe Street, Baltimore, MD 21205, USA; yyang279@jh.edu; 5Department of Mechanical Engineering, City University of Hong Kong, Hong Kong 999077, China

**Keywords:** dietary patterns, stair climbing, major adverse cardiovascular events, UK Biobank

## Abstract

Background: The protective effect of a healthy diet combined with stair climbing on cardiovascular health is unclear. We aimed to assess the independent and joint associations of dietary patterns and stair climbing with major adverse cardiovascular events (MACEs). Methods: We included 117,384 participants with information on dietary intake and stair climbing from the UK Biobank (UKBB). We excluded participants with MACEs at baseline and death within two years of follow-up. We used restricted cubic spline (RCS) plots describing the linear or nonlinear associations between dietary patterns (the alternate Mediterranean diet score (AMED), dietary approaches to stop hypertension (DASH), the healthful planted-based diet index (HPDI) and the alternate healthy eating index-2010 (AHEI-2010)) and stair climbing and MACEs. COX regressions estimated the hazard ratios (HRs) for incident MACEs associated with dietary patterns combined with stair climbing, and adjusted for sociodemographic, lifestyle and medical factors. Results: The UKBB documented 9408 MACEs over a median follow-up of 13.3 years. Four dietary patterns were negatively and linearly associated with MACEs (P-nonlinear > 0.05), whereas daily stair climbing was negatively and nonlinearly associated with MACEs (P-nonlinear = 0.011). All of the dietary patterns had significant multiplicative interactions with stair climbing (all *p*-values < 0.05). The three dietary patterns had the lowest risk ratios for MACEs in the highest tertile (T3) combined with daily stair climbing of 60–100 steps (AMED: 0.78 (0.68, 0.89), DASH: 0.80 (0.70, 0.91) and HPDI: 0.86 (0.75, 0.98)), whereas the AHEI-2010 had the lowest HRs for MACEs in the T1 combined with stair climbing of 110–150 steps (AHEI-2010: 0.81 (0.71, 0.93)). Conclusions: Maintaining healthy dietary patterns and adhering to an average of 60–100 steps of stair climbing per day at home can be effective in preventing cardiovascular health-related events.

## 1. Introduction

Cardiovascular disease (CVD) ranks among the leading causes of global disease burden, constituting the predominant etiology of premature mortality, incurring significant healthcare expenditures and diminishing health-related quality of life [1,2,3]. From 1990 to 2019, the total number of CVD cases worldwide grew from 271 million to 523 million. The prevalence of CVD is expected to increase by 90% by 2050 [4,5]. The progression of CVD encompasses a spectrum of conditions affecting the heart and vasculature, including atherosclerosis, hypertension, coronary artery disease and heart failure [6]. These pathological symptoms cause major adverse cardiovascular events (MACEs). MACEs are delineated as incidences of nonfatal myocardial infarction, stroke and fatal cardiovascular events [7]. Health authorities recommend a healthy lifestyle, including both a quality diet and regular physical activity, for the prevention of CVD [8]. Thus, comprehending the relationship between a healthy lifestyle and CVD-induced MACEs may significantly enhance the efficacy of intervention strategies for CVD patients.

Vigorous physical activity is correlated with more favorable health outcomes, which is attributable to the robust cardioprotective adaptations elicited by the increased levels of physical exertion. Intermittent non-exercise physical activity for 5–10 min decreased the risk of MACEs by 41% [2]. While vigorous physical activity represents a time-efficient modality for attaining health goals, exercise regimens characterized by high intensity, such as high-intensity interval training, exhibit low participation rates among middle-aged and older adults [9,10]. In addition, the scarcity of time due to work and family responsibility is deemed as another predominant obstacle to physical activity. Stair climbing constitutes a convenient and readily accessible method for integrating physical activity into everyday routines [11]. Participants who climbed stairs 16–20 times/day had a 23% lower risk of atherosclerotic cardiovascular disease than those who climbed stairs 0 times/day [12]. The anticipated health advantages of stair climbing encompass augmented cardiovascular fitness, optimized lipoprotein profiles, reduced body adiposity, diminished fasting blood glucose levels and augmented lower-limb strength [13,14]. Thus, exploring the beneficial effects of stair climbing on CVD is essential.

As poor diet quality is strongly associated with the elevated risk of cardiovascular disease morbidity and mortality, diet is another critical lifestyle determinant for chronic diseases and represents a feasible target for the prevention and postponement of CVD [15,16]. Empirical evidence from prior research has substantiated that the adherence to healthy dietary patterns was associated with a low risk of CVD, such that the Mediterranean diet (AMED), alternative healthy diet index (AHEI) and healthy plant-based diet index (HPDI) were associated with a 17%, 21% and 14% lower risk of CVD at the highest quintile compared to the lowest quintile, respectively [17]. In addition, the higher the compliance with the dietary approaches to stop hypertension (DASH) diet, the lower the risk of CVD [18]. Thus, it is important to comprehensively explore the effects of different dietary patterns on CVD risks.

To date, only a few studies have investigated the interactive or combined associations between dietary patterns and stair climbing, with a limited subset having explored CVD outcomes. To investigate the cardiovascular protective effects of these two healthy lifestyles, in this study, we investigated the individual and combined effects of four healthy dietary patterns (the AMED, AHEI, HPDI and DASH) and daily stair climbing frequency on MACEs in a large prospective cohort study.

## 2. Materials and Methods

### 2.1. Study Population

The UK Biobank (UKBB) was a population-based cohort of more than 500,000 participants aged 39 to 70 years at enrollment between 2006 and 2010. These participants were recruited from 1 of the 22 assessment centers across England, Scotland and Wales, where they completed a self-administered touch-screen questionnaire and face-to-face interview. Details of the study design have been previously documented [19]. The UKBB study’s ethical approval was granted by the National Information Governance Board for Health and Social Care and the NHS North West Multicenter Research Ethics Committee. All participants provided written informed consent at recruitment.

Individuals who completed ≥2 dietary assessments were included in the analysis, after further sequentially excluding individuals with an implausible daily total calorie intake, with CVD at baseline and those died within the first 2 years (Appendix A).

### 2.2. Ascertainment of Diseases

MACEs are defined as incidences of nonfatal myocardial infarction, stroke and fatal cardiovascular events. The inpatient hospital data were available since 1997 in the UKBB [20]. Incident CVD was identified according to the International Classification of Diseases-10th revision codes I21-I25, I50, I60, I61, I63 and I64. Participants were followed from the date of attending the assessment centers until the date of the CVD diagnosis, the loss to follow-up, death or the end of the follow-up (31 October 2022 for centers in England, 31 August 2022 for centers in Scotland and 31 May 2022 for centers in Wales), whichever came first.

### 2.3. Definition of Lifestyle Factors

Diet was assessed using a web-based 24 h dietary assessment tool (the validated Oxford WebQ). A sub-cohort of the UK Biobank completed the assessment on ≥1 of the 5 occasions between April 2009 and June 2012. The Oxford WebQ has been validated against an interviewer-administered 24 h dietary recall, producing a mean Spearman correlation coefficient for macronutrients of 0.62 (range 0.54–0.69) [21].

We computed the amount of each food consumed by multiplying the assigned portion size by the quantity consumed. The nutrient amounts were computed by multiplying the quantity of each food consumed by the nutrient content of the portion (McCance and Widdowson’s The Composition of Foods and its Supplements) and then summing this across all food groups. To estimate the usual intake, the average food/nutrient intakes of the two or more dietary assessments were used in the analysis.

#### 2.3.1. Alternate Mediterranean Diet Score (AMED)

The AMED score was calculated based on 9 food/nutrient groups. For whole grains, vegetables (excluding potatoes), fruits, nuts and seeds, legumes, fish and the ratio of monounsaturated fatty acid to saturated fatty acid, intakes above the sex-specific median of the included participants were given 1 point, and all other intakes were given 0 points. For meat (red and processed meats), 0 points were assigned to individuals above the median intake, and 1 point was assigned to all others. Alcohol intake between 5 and 15 g/d was assigned 1 point. The total AMED score ranged from 0 to 9, with a higher score representing a healthier diet [22].

#### 2.3.2. Alternate Healthy Eating Index-2010 (AHEI-2010)

The AHEI-2010 score in our analysis was calculated using the following 10 food and nutrient components: whole grains, vegetables, fruits, nuts and legumes, sugar-sweetened beverages and fruit juice, red/processed meat, fish (long-chain *n*-3 fat substitution), polyunsaturated fatty acids, added salt (sodium substitution) and alcohol. Each component was scored proportionally on a scale from 0 to 10, yielding a total AHEI-2010 score ranging from 0 to 100. Trans fatty acids were excluded from the calculation due to unavailability in the dataset.

#### 2.3.3. Dietary Approaches to Stop Hypertension (DASH)

The DASH dietary pattern score focused a priori on its eight key food or nutrient group components: a high intake of fruits, vegetables, nuts and legumes, low-fat dairy products and whole grains, and a low intake of sodium, sweetened beverages and red and processed meats. For each of the components, all of the participants were classified into fifths according to their intake ranking. The fifth rankings were assigned as the component score for each of the five healthy food group items (fruits, vegetables, nuts and legumes, low-fat dairy products and whole grains). For example, the first fifth was assigned 1 point and the last fifth was assigned 5 points. In contrast, a low intake is desired for sodium, sweetened beverages and red and processed meats. Therefore, the lowest fifth for each of these three items was given a score of 5 points and the highest fifth was given 1 point. We then summed the component scores to obtain an overall DASH score for each participant, ranging from 8 to 40. This DASH score has been successfully used in prior cohort studies for the risk of cardiovascular disease and kidney stones [23].

#### 2.3.4. Healthful Plant-Based Diet Index (HPDI)

The HPDI score was calculated using 17 food groups, excluding vegetable oil due to its unavailability in the UKBB. Each food group was assigned a score from 1 to 5 based on quintiles. For plant-based foods (whole grains, fruits, vegetables, nuts, legumes and tea/coffee), the highest quintile received a score of 5 and the lowest quintile received a score of 1. For animal-based foods (animal fat, dairy, eggs, fish/seafood, meat and other animal-based foods), the highest quintile was assigned a score of 1 and the lowest quintile was assigned a score of 5. For other groups (refined grains, potatoes, sugary drinks, fruit juices and sweets/desserts), the highest quintile received a score of 1. The total HPDI score ranged from 17 to 85, with higher scores indicating a healthier diet [24].

### 2.4. Assessment of Covariates

We obtained information on age, sex, race/ethnicity, education, the Townsend deprivation index, smoking, drinking, total energy intake, physical activity, sleep duration, body mass index (BMI), systolic blood pressure, diastolic blood pressure and baseline diabetes through questionnaires on a touch-screen computer. Age, sex, race/ethnicity, education, smoking status and alcohol intake were self-reported. The Townsend area deprivation index was derived from the postal code of the residence using aggregated data on unemployment, car and home ownership and household overcrowding [25]. Sleep duration was assessed based on the question “About how many hours’ sleep do you get in every 24 h?” Physical activity was assessed using a short form of the International Physical Activity Questionnaire. Height was measured to the nearest centimeter using a stadiometer (Seca 202; Hamburg, Germany) and body weight was measured to the nearest 0.1 kg using a body composition analyzer (Tanita BC-418; Tokyo, Japan). BMI was calculated as weight (kg)/height (m)^2^. Only the assessment at baseline was used in this study.

### 2.5. Statistical Analysis

Baseline characteristics were described according to the frequency of daily stair climbing (none, 10–50 steps/day, 60–100 steps/day, 110–150 steps/day or >150 steps/day). Cox proportional hazard regression models were used to examine the associations of each of the 4 dietary scores with the incidence of CVD and adjusted for potential confounding variables. A series of potential confounders were identified based on the current literature [16]. The models were adjusted for age, sex, race/ethnicity, education and the Townsend deprivation index (model 1), as well as drinking, smoking, total energy intake, physical activity, sleep duration, BMI, systolic blood pressure, diastolic blood pressure and baseline diabetes (model 2). Model 2 was the primary model. The proportional hazards assumption was met by using the Kolmogorov-type supremum test [26]. Hazard ratios (HRs) with 95% confidence intervals (CIs) were presented. Multiple imputations by chained equations were utilized to address missing data [27].

We explored whether or not daily stair climbing is a potential effect modifier for healthy diet pattern and MACEs. To test the multiplicative interaction between daily stair climbing and a healthy diet score, we treated the group coded “none” for daily stair climbing and a low healthy diet score (the lowest tertile of the healthy diet score) as the reference group, and calculated HRs for the other groups. The *p*-value for the interaction was estimated using the joint test. We further examined the association between daily stair climbing and MACEs as stratified by the tertiles of the healthy diet score. Statistical analyses were conducted with R, version 4.2.1 (R Foundation for Statistical Computing, Vienna, Austria). A 2-sided *p* < 0.05 was considered statistically significant.

## 3. Results

### 3.1. Baseline Characteristics of Participants in the UKBB

Of the 117,384 participants from the UKBB, the mean age was 55.9 ± 7.8 years, with 42.6% male and 96.7% White (Table 1). Divided into five categories according to daily steps of stair climbing, participants who took more than 150 steps per day had the lowest BMI (25.7 kg/m^2^), prevalence of diabetes (2.4%), mean systolic (136.0 mmHg) and diastolic blood pressure (81.4 mmHg), and the highest mean total energy intake (2074.3 kcal) and four healthy dietary pattern scores in each category. There were significant differences in the education level, the Townsend deprivation index, physical activity, sleep duration, smoking status and drinking status among all categories of participants (*p*-value < 0.001).

### 3.2. Independent Associations of Frequency of Stair Climbing and Four Healthy Dietary Patterns with MACEs

Restricted cubic spline (RCS) analyses assessed the associations between the frequency of stair climbing at home and of the four dietary pattern scores with MACEs (Figure 1 and Figure 2). The HRs for MACEs decreased significantly with an increasing frequency of stair climbing compared to no stair climbing (P for overall < 0.001), and this association was nonlinear (P for nonlinear = 0.011). The lowest risk of MACEs was observed in climbing 110–150 steps compared with the other groups. Except for the AHEI-2010 (P for overall = 0.145), the other three healthy dietary patterns were linearly and significantly negatively associated with the HRs for MACEs (P for overall < 0.001; AMED: P for nonlinear = 0.206; DASH: P for nonlinear = 0.942; HPDI: P for nonlinear = 0.973).

### 3.3. Joint Associations of Frequency of Stair Climbing and Four Healthy Dietary Patterns with MACEs

We observed a statistically significant multiplicative interaction between stair climbing and each of the healthy dietary patterns, suggesting that the frequency of stair climbing appears to have a synergistic or antagonistic effect with dietary pattern scores on the incidence of MACEs (Table 2). In model 1, participants climbing 60–100 steps per day minimized the HRs for the incidence of MACEs at the highest tertile (T3) of the three dietary patterns, the AMED, DASH and HPDI (HR (95%CI): 0.72 (0.63–0.82); HR (95%CI): 0.74 (0.65–0.84); HR (95%CI): 0.79 (0.70–0.90)). At the T3 of the AHEI-2010, participants climbing 110–150 steps per day decreased the risk ratio for MACEs by 25% compared to the reference population. We observed that the largest percentage of all participants climbed 60–100 steps per day (Table 1 and Table 2). In this group, the higher were the four dietary pattern scores, the lower were the HRs for the incidence of MACEs (T1-T3: AMED: 17%; AHEI-2010: 9%; DASH: 11%; HPDI: 12%). In addition, we found that increasing the frequency of stair climbing offset the increased HRs due to the low dietary quality. Compared with the participants with dietary pattern scores at the T2 or T3 tertile who climbed less than 50 steps (none and 10–50 steps) per day, participants at the T1 tertile who climbed 110–150 steps per day achieved the same or even lower HRs for the incidence of MACEs. The findings of model 2 were generally consistent with those of model 1. We additionally adjusted for the key confounders of hypertension, hypercholesterolemia, obesity (BMI > 30 kg/m^2^), atherosclerotic disease and the history of cardiovascular disease, and the results are displayed in Appendix A.

### 3.4. Joint Associations of Frequency of Stair Climbing and Four Healthy Dietary Patterns with MACEs in the Obesity Subgroup

To further investigate the effects of healthy dietary patterns and stair climbing on the risk of MACEs in people with basal metabolic abnormalities, we defined obese and non-obese according to BMI (Figure 3 and Figure 4). We observed no significant association between healthy dietary patterns and MACEs in participants who did not climb stairs per day. In the obese group (BMI ≥ 30 kg/m^2^), the combination of the AHEI-2010, AMED and DASH with stair climbing of more than 60 steps per day reduced the HRs for the incidence of MACEs to varying degrees, while the combination of the HPDI and stair climbing was not significant. We found that the combined intervention of healthy dietary patterns and stair climbing in the participants without obesity did not reduce the HRs for the incidence of MACEs as much as in the participants with obesity. Except for the AHEI-2010, the other three dietary patterns were able to significantly reduce the HRs for the incidence of MACEs by combining stairs with more than 60 steps of stair climbing per day only at the T3.

## 4. Discussion

This is the first large prospective cohort-based study designed to assess the association between four healthy dietary patterns combined with stair climbing per day and MACEs. We observed a significant multiplicative interaction between dietary patterns and stair climbing, and their joint effect significantly reduced the HRs for the incidence of MACEs.

Our findings are consistent with previous ones that the AMED, DASH, AHEI-2010 and HPDI are protective against the incidence of MACEs, although the association between the AHEI-2010 and the HRs for the incidence of MACEs was not statistically significant in the RCS analysis. The association between a healthy diet and lower CVD events has been reported in several large cohorts [28,29]. A previous UKBB study showed that participants in the highest quintile of the AMED, HPDI and AHEI scores had 17%, 14% and 21% lower HRs for CVD compared with the lowest quintile, respectively [16]. In addition, the Nurses’ Health Study (NHS), NHS II and the Health Professionals Follow-up Study (HPFS), three cohorts with up to 32 years of follow-up, also reported that higher-quality diets (AMED, AHEI and HPDI) were associated with an 8–21% reduction in HRs for CVD [17]. Of these dietary patterns, only the highest tertiles of the AMED significantly reduced an individual’s HR for the incidence of MACEs by 17% compared with the lowest tertiles, independent of stair climbing. A cohort study of participants with coronary heart disease from 39 countries showed that higher adherence to the Mediterranean diet was associated with a lower risk of MACEs [30]. Promoting the intake of whole grains, fish and monounsaturated fatty acid-rich foods contributes to cardiovascular health in healthy people or patients with obesity.

High adherence to healthy dietary patterns such as the AMED or DASH reduces systemic inflammation levels, improves metabolic disorders (obesity and diabetes) and thus reduces the occurrence of more severe MACEs [31,32,33,34]. These healthy diets promote the intake of essential micronutrients and bioactives that not only improve cardiovascular and cerebrovascular metabolic health, in addition to reducing cellular oxidative stress, but also help to slow down the aging process [35]. These findings emphasize the public health importance of a high degree of adherence to healthy dietary patterns, reducing the huge economic burden of disease and greatly improving quality of life.

Previous studies have confirmed that physical activities (running, cycling, climbing stairs at home and others) significantly reduce the risk of cardiovascular disease [2,11]. Of these activities, stair climbing has received increasing attention for not requiring cost, equipment or skill [11]. Our observations are consistent with existing evidence supporting the positive impact of stair climbing on CVD. A prior study showed that frequent stair climbing reduced the risk of the incidence of atrial fibrillation in urban Japanese [36,37], and a previous study by UKBB found that participants who regularly climbed stairs at home had a 5% to 14% lower risk of developing T2D [11]. Interestingly, the frequency of stair climbing was not linearly associated with a reduced risk of developing MACEs. Participants who climbed 110–150 steps per day were better protected against T2D and MACEs than those who climbed less than 110 steps or more than 150 steps per day [11].

For the first time, we found a significant interaction between healthy dietary patterns and stair climbing, and observed that the two appeared to synergize to protect cardiovascular health. Although the biological mechanisms of this interaction are unknown, this conclusion is reasonable in terms of biological associations. All four of the healthy diets have been shown to be effective in reducing the risk of metabolic syndrome [16]. Stair climbing, a combination of aerobic and resistance exercise, has been shown to be effective in reducing abdominal obesity, lowering body weight and improving insulin resistance and lipid profiles [14,37,38,39]. The combination of healthy lifestyles therefore helps to improve the metabolism and thus reduce the incidence of cardiovascular disease. An Australian cohort study of over 85,000 older adults found that high-quality dietary scores and ≥300 min/week of moderate-to-vigorous physical activity (MVPA) reduced the risk of CVD mortality by 40–50%, but did not find a significant multiplicative interaction [6]. We suggest that this may be related to the older age of the participants, the shorter duration of the follow-up and the limited exercise modalities.

We observed that the AMED, DASH and HPDI scores at the highest tertile combined with 60–100 steps of stair climbing per day minimized the risk of MACEs, with HRs of 0.78 (0.68–0.89), 0.80 (0.70–0.91) and 0.86 (0.75–0.98), respectively. Climbing 60–100 steps per day accounted for the largest proportion of all participants, and, when combined with a high-quality diet, maximized cardiovascular health. Whole grain, vegetable and fruit intake are recommended in all four healthy diets, with the DASH emphasizing a low-salt and low-sodium diet to prevent hypertension [40], and the HPDI emphasizing a plant-based diet to reduce meat intake [16]. Compared with the AMED, fish intake (including freshwater fish and seafood) may be a key factor influencing the incidence of MACEs. Previous studies have found that fish intake is associated with a low cardiovascular risk [41,42]. Therefore, an increased fish intake and the adherence to the AMED dietary pattern is recommended for the prevention of cardiovascular disease.

Our study has several strengths. The large sample of more than 110,000 people with a median of 13.3 years of follow-up provided sufficient power to assess the combined effect of dietary patterns and stair climbing on the risk of developing MACEs. Secondly, stair climbing is an exercise that is easy to perform in life, does not require any equipment or skills and costs nothing compared with other ball games or cycling. There is an interaction between healthy dietary patterns and stair climbing to jointly reduce the incidence of MACEs, and we provided the optimal frequency of stair climbing per day. Thirdly, we calculated four important healthy dietary patterns and suggested different healthy dietary patterns for obese and non-obese groups in combination with daily stair climbing to achieve optimal results. However, there are some limitations of this study. Firstly, the frequency of stair climbing was obtained by a questionnaire rather than based on professional activity logger measurements, which may be subject to some reporting bias. Similarly, the calculation of dietary pattern scores was based on a 24-h diet questionnaire. Although the web-based 24-h dietary assessment tool used in the UKBB study has been validated against biomarkers, the presence of self-reported bias is unavoidable. Finally, the results of observational studies cannot establish causality. Therefore, more randomized controlled clinical trials are needed in the future to further confirm our conclusions.

## 5. Conclusions

In conclusion, the adherence to healthy dietary patterns and combined with daily stair climbing is associated with a reduced risk of MACEs. Our results suggest that attention should be paid to the combined effects of these two lifestyles, dietary quality and low-cost exercise. The optimal combination should be chosen to prevent MACEs in obese and non-obese groups, respectively.

## Figures and Tables

**Figure 1 nutrients-16-03576-f001:**
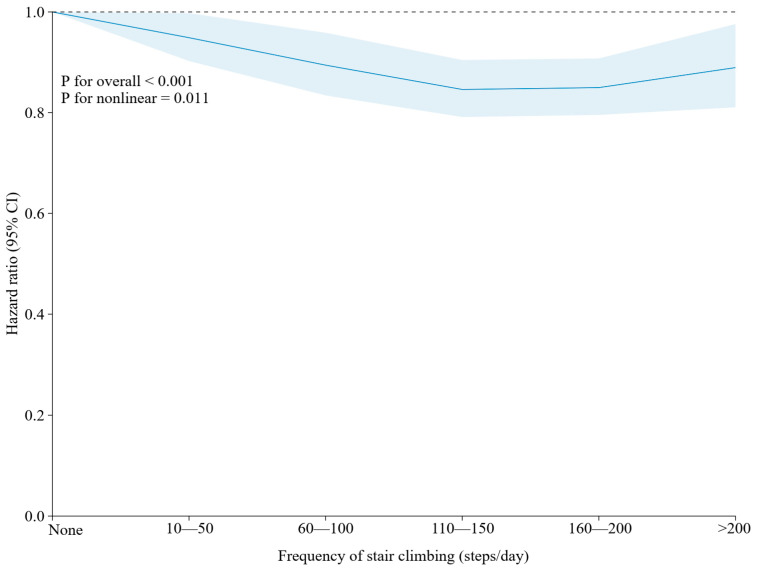
Restricted cubic spline (RCS) plots of the association between the frequency of stair climbing and MACEs. Note: Adjusted for sex, age, the Townsend deprivation index and education levels. *p* < 0.05 was considered significant.

**Figure 2 nutrients-16-03576-f002:**
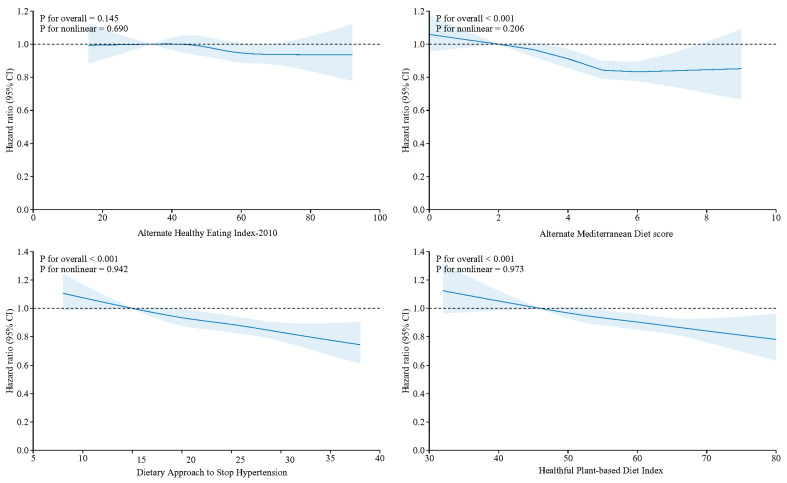
Restricted cubic spline (RCS) plots of the association between the four dietary patterns and MACEs. Note: Adjusted for sex, age, the Townsend deprivation index and education levels. *p* < 0.05 was considered significant.

**Figure 3 nutrients-16-03576-f003:**
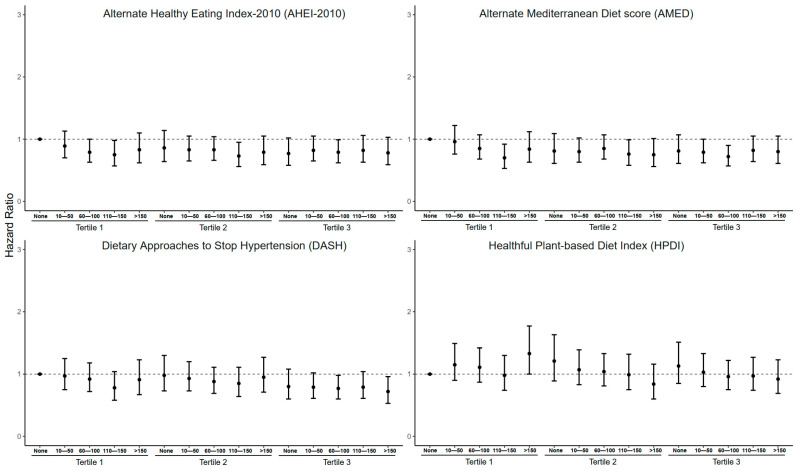
Association between dietary patterns combined with stair climbing and MACEs in the obesity group (BMI ≥ 30 kg/m^2^). Note: The model was adjusted for age, sex, race/ethnicity, education levels, the Townsend deprivation index, drinking, smoking, total energy intake, physical activity, sleep duration, BMI, systolic blood pressure, diastolic blood pressure and baseline diabetes. Obesity: BMI ≥ 30.

**Figure 4 nutrients-16-03576-f004:**
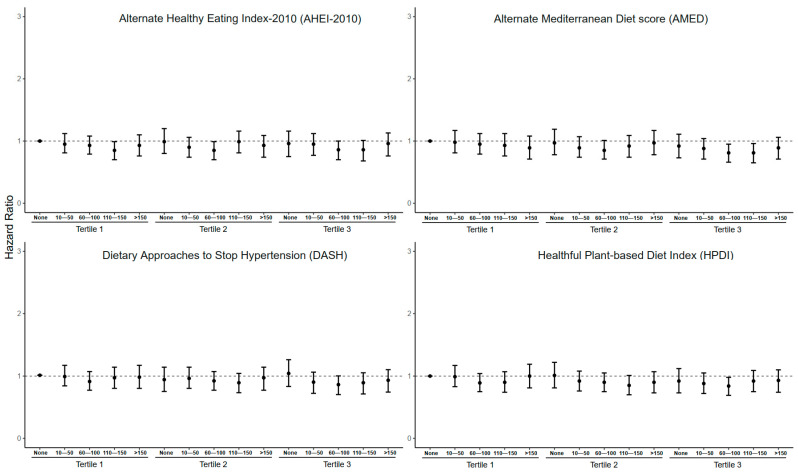
Association between dietary patterns combined with stair climbing and MACEs in the non-obesity group (BMI < 30 kg/m^2^). Note: The model was adjusted for age, sex, race/ethnicity, education levels, the Townsend deprivation index, drinking, smoking, total energy intake, physical activity, sleep duration, BMI, systolic blood pressure, diastolic blood pressure and baseline diabetes. Non-obesity: BMI < 30.

**Table 1 nutrients-16-03576-t001:** Baseline characteristics of the 117,384 participants in UK Biobank by daily stair climbing.

Variable	Frequency of Stair Climbing (Steps/Day)							
	Overall (*n* = 117,384)	None (*n* = 8745)	10–50 (*n* = 21,277)	60–100 (*n* = 43,624)	110–150 (*n* = 24,126)	>150 (*n* = 19,612)	*p*-Value	*p* for Trend
Age, year, and mean (SD)	55.9 (7.8)	57.9 (7.8)	55.2 (7.7)	55.8 (7.7)	55.9 (7.9)	55.8 (7.9)	<0.001	<0.001
Sex							<0.001	<0.001
Men	50,033 (42.6%)	3733 (42.7%)	9505 (44.7%)	18,933 (43.4%)	10,039 (41.6%)	7823 (39.9%)		
Women	67,351 (57.4%)	5012 (57.3%)	11,772 (55.3%)	24,691 (56.6%)	14,087 (58.4%)	11,789 (60.1%)		
Race/ethnicity							<0.001	<0.001
White	113,472 (96.7%)	8464 (96.8%)	20,281 (95.3%)	42,339 (97.1%)	23,428 (97.1%)	18,960 (96.7%)		
Black	906 (0.8%)	81 (0.9%)	244 (1.1%)	300 (0.7%)	136 (0.6%)	145 (0.7%)		
Asian	1332 (1.1%)	70 (0.8%)	351 (1.6%)	463 (1.1%)	237 (1.0%)	211 (1.1%)		
Other	1674 (1.4%)	130 (1.5%)	401 (1.9%)	522 (1.2%)	325 (1.3%)	296 (1.5%)		
Education level							<0.001	<0.001
A/AS levels or equivalent (11–12 years)	15,899 (13.5%)	1059 (12.1%)	2879 (13.5%)	5928 (13.6%)	3362 (13.9%)	2671 (13.6%)		
O levels/secondary or equivlanet (7–10 years)	26,985 (23.0%)	2379 (27.2%)	5102 (24.0%)	10,245 (23.5%)	5186 (21.5%)	4073 (20.8%)		
University	55,976 (47.7%)	3222 (36.8%)	9572 (45.0%)	20,673 (47.4%)	12,208 (50.6%)	10,301 (52.5%)		
Vocational	11,156 (9.5%)	1037 (11.9%)	2171 (10.2%)	4158 (9.5%)	2209 (9.2%)	1581 (8.1%)		
Other	7368 (6.3%)	1048 (12.0%)	1553 (7.3%)	2620 (6.0%)	1161 (4.8%)	986 (5.0%)		
Townsend deprivation index	−1.7 (2.8)	−1.3 (3.2)	−1.1 (3.1)	−1.8 (2.7)	−1.9 (2.6)	−1.8 (2.7)	<0.001	<0.001
Body mass index, kg/m^2^	26.6 (4.5)	27.2 (4.9)	27.5 (5.1)	26.7 (4.5)	26.1 (4.1)	25.7 (4.0)	<0.001	<0.001
Physical activity, MET-minutes/week	2258.1 (2274.3)	2234.3 (2296.1)	1944.7 (2132.2)	2139.8 (2157.9)	2353.0 (2268.1)	2755.0 (2568.2)	<0.001	<0.001
Sleep duration, hour	7.2 (1.0)	7.2 (1.2)	7.1 (1.1)	7.2 (1.0)	7.2 (1.0)	7.1 (1.0)	<0.001	0.244
Smoking status							<0.001	<0.001
Current	7946 (6.8%)	731 (8.4%)	1748 (8.2%)	2836 (6.5%)	1450 (6.0%)	1181 (6.0%)		
Former	41,097 (35.0%)	3336 (38.1%)	7659 (36.0%)	15,191 (34.8%)	8416 (34.9%)	6495 (33.1%)		
Never	68,341 (58.2%)	4678 (53.5%)	11,870 (55.8%)	25,597 (58.7%)	14,260 (59.1%)	11,936 (60.9%)		
Drinking status							<0.001	<0.001
Current	110,843 (94.4%)	8052 (92.1%)	19,790 (93.0%)	41,404 (94.9%)	23,001 (95.3%)	18,596 (94.8%)		
Former	3226 (2.7%)	347 (4.0%)	743 (3.5%)	1096 (2.5%)	557 (2.3%)	483 (2.5%)		
Never	3315 (2.8%)	346 (4.0%)	744 (3.5%)	1124 (2.6%)	568 (2.4%)	533 (2.7%)		
Mean total energy intake, kcal	2057.1 (468.7)	2023.8 (470.9)	2035.5 (473.2)	2060.3 (465.7)	2068.7 (461.9)	2074.3 (476.0)	<0.001	<0.001
Systolic blood pressure, mm Hg	136.5 (18.2)	138.4 (18.1)	136.2 (17.9)	136.5 (18.1)	136.2 (18.3)	136.0 (18.4)	<0.001	<0.001
Diastolic blood pressure, mm Hg	81.7 (9.9)	82.1 (9.9)	82.1 (9.9)	81.8 (9.9)	81.4 (9.9)	81.4 (9.9)	<0.001	<0.001
AHEI-2010	53.0 (11.7)	53.0 (11.7)	52.1 (11.8)	52.7 (11.6)	53.3 (11.6)	53.9 (11.7)	<0.001	<0.001
AMED	4.2 (1.3)	4.2 (1.3)	4.1 (1.3)	4.2 (1.2)	4.3 (1.2)	4.3 (1.2)	<0.001	<0.001
HPDI	56.8 (6.5)	56.8 (6.6)	56.2 (6.6)	56.6 (6.5)	57.1 (6.4)	57.4 (6.5)	<0.001	<0.001
DASH	22.8 (4.6)	22.9 (4.6)	22.4 (4.6)	22.7 (4.5)	22.9 (4.5)	23.2 (4.6)	<0.001	<0.001
Diabetes							<0.001	<0.001
No	113,559 (96.7%)	8337 (95.3%)	20,374 (95.8%)	42,227 (96.8%)	23,475 (97.3%)	19,146 (97.6%)		
Yes	3825 (3.3%)	408 (4.7%)	903 (4.2%)	1397 (3.2%)	651 (2.7%)	466 (2.4%)		

Note: Descriptive data are shown as the mean (SD), while categorical variables are reported as *n* (%). A *p*-value < 0.05 is considered significant. Abbreviations: SD = standard deviation; AHEI-2010 = alternate healthy eating index-2010; AMED = alternate Mediterranean diet score; DASH = dietary approach to stop hypertension; HPDI = healthful plant-based diet index; MET = metabolic equivalent.

**Table 2 nutrients-16-03576-t002:** Association between dietary patterns combined with stair climbing and MACEs.

	MACEs					
Dietary Patterns and Climbing Stairs		Model 1		Model 2		
	Case/N	HR (95% CI)	*p* Value	HR (95% CI)	*p*-Value	*p* for Interaction
**Alternate Mediterranean Diet score (AMED)**						
AMED T1 (0–3)						0.017
Stair climbing (steps/day) None	271/2389	Reference		Reference		
Stair climbing (steps/day) 10–50	577/6312	0.98 (0.85–1.14)	0.829	0.98 (0.84–1.13)	0.733	
Stair climbing (steps/day) 60–100	970/11,523	0.89 (0.78–1.02)	0.092	0.91 (0.80–1.05)	0.190	
Stair climbing (steps/day) 110–150	463/6012	0.82 (0.70–0.95)	0.009	0.86 (0.74–1.00)	0.049	
Stair climbing (steps/day) > 150	358/4725	0.81 (0.69–0.94)	0.007	0.87 (0.74–1.02)	0.086	
AMED T2 (4–5)						
Stair climbing (steps/day) None	276/2598	0.90 (0.76–1.06)	0.203	0.91 (0.77–1.08)	0.298	
Stair climbing (steps/day) 10–50	545/6452	0.86 (0.74–0.99)	0.038	0.86 (0.74–0.99)	0.042	
Stair climbing (steps/day) 60–100	1052/13,064	0.81 (0.71–0.93)	0.002	0.85 (0.74–0.97)	0.016	
Stair climbing (steps/day) 110–150	573/7168	0.80 (0.70–0.93)	0.003	0.87 (0.75–1.00)	0.055	
Stair climbing (steps/day) > 150	443/5625	0.83 (0.71–0.96)	0.016	0.91 (0.78–1.06)	0.220	
AMED T3 (6–9)						
Stair climbing (steps/day) None	369/3758	0.83 (0.71–0.97)	0.020	0.88 (0.75–1.03)	0.105	
Stair climbing (steps/day) 10–50	668/8513	0.81 (0.70–0.93)	0.003	0.84 (0.73–0.97)	0.017	
Stair climbing (steps/day) 60–100	1370/19,037	0.72 (0.63–0.82)	<0.001	0.78 (0.68–0.89)	<0.001	
Stair climbing (steps/day) 110–150	789/10,946	0.72 (0.63–0.83)	<0.001	0.80 (0.69–0.92)	0.002	
Stair climbing (steps/day) > 150	684/9262	0.76 (0.66–0.87)	<0.001	0.86 (0.74–0.99)	0.032	
**Alternate Healthy Eating Index-2010 (AHEI-2010)**						
AHEI-2010 T1 (0–47)						<0.001
Stair climbing (steps/day) None	340/2847	Reference		Reference		
Stair climbing (steps/day) 10–50	700/7537	0.95 (0.84–1.09)	0.479	0.94 (0.83–1.08)	0.386	
Stair climbing (steps/day) 60–100	1251/14,597	0.87 (0.77–0.98)	0.021	0.89 (0.79–1.00)	0.048	
Stair climbing (steps/day) 110–150	585/7574	0.78 (0.68–0.89)	<0.001	0.81 (0.71–0.93)	0.002	
Stair climbing (steps/day) > 150	476/5866	0.83 (0.73–0.96)	0.011	0.90 (0.78–1.03)	0.127	
AHEI-2010 T2 (48–58)						
Stair climbing (steps/day) None	292/2833	0.89 (0.76–1.05)	0.162	0.91 (0.78–1.07)	0.246	
Stair climbing (steps/day) 10–50	551/6863	0.86 (0.75–0.98)	0.025	0.86 (0.75–0.99)	0.031	
Stair climbing (steps/day) 60–100	1089/14,275	0.80 (0.70–0.90)	<0.001	0.84 (0.74–0.95)	0.005	
Stair climbing (steps/day) 110–150	645/7945	0.85 (0.75–0.97)	0.018	0.92 (0.80–1.05)	0.208	
Stair climbing (steps/day) > 150	480/6318	0.81 (0.70–0.93)	0.003	0.89 (0.77–1.02)	0.096	
AHEI-2010 T3 (59–95)						
Stair climbing (steps/day) None	284/3065	0.87 (0.74–1.02)	0.085	0.91 (0.78–1.07)	0.264	
Stair climbing (steps/day) 10–50	539/6877	0.88 (0.77–1.01)	0.067	0.91 (0.79–1.04)	0.153	
Stair climbing (steps/day) 60–100	1052/14,752	0.78 (0.69–0.88)	<0.001	0.84 (0.74–0.95)	0.005	
Stair climbing (steps/day) 110–150	595/8607	0.75 (0.66–0.86)	<0.001	0.84 (0.73–0.96)	0.009	
Stair climbing (steps/day) > 150	529/7428	0.80 (0.70–0.92)	0.002	0.91 (0.79–1.04)	0.157	
**Dietary Approaches to Stop Hypertension (DASH)**						
DASH T1 (0–20)						0.004
Stair climbing (steps/day) None	320/2743	Reference		Reference		
Stair climbing (steps/day) 10–50	661/7429	0.96 (0.84–1.09)	0.513	0.95 (0.83–1.08)	0.418	
Stair climbing (steps/day) 60–100	1142/13,976	0.85 (0.75–0.96)	0.011	0.88 (0.77–0.99)	0.038	
Stair climbing (steps/day) 110–150	581/7239	0.84 (0.74–0.97)	0.015	0.89 (0.78–1.02)	0.097	
Stair climbing (steps/day) > 150	444/5501	0.87 (0.75–1.00)	0.051	0.94 (0.81–1.08)	0.375	
DASH T2 (21–25)						
Stair climbing (steps/day) None	284/2834	0.84 (0.72–0.99)	0.039	0.88 (0.75–1.03)	0.111	
Stair climbing (steps/day) 10–50	620/7025	0.90 (0.79–1.03)	0.129	0.92 (0.80–1.05)	0.210	
Stair climbing (steps/day) 60–100	1192/14,488	0.83 (0.73–0.93)	0.002	0.88 (0.77–0.99)	0.036	
Stair climbing (steps/day) 110–150	609/8050	0.76 (0.66–0.87)	<0.001	0.83 (0.72–0.95)	0.007	
Stair climbing (steps/day) > 150	507/6424	0.82 (0.71–0.94)	0.005	0.91 (0.79–1.04)	0.168	
DASH T3 (26–40)						
Stair climbing (steps/day) None	312/3168	0.89 (0.76–1.04)	0.138	0.94 (0.80–1.10)	0.413	
Stair climbing (steps/day) 10–50	509/6823	0.80 (0.70–0.92)	0.002	0.83 (0.72–0.96)	0.011	
Stair climbing (steps/day) 60–100	1058/15,160	0.74 (0.65–0.84)	<0.001	0.80 (0.70–0.91)	0.001	
Stair climbing (steps/day) 110–150	635/8837	0.75 (0.66–0.86)	<0.001	0.84 (0.73–0.96)	0.010	
Stair climbing (steps/day) > 150	534/7687	0.75 (0.65–0.86)	<0.001	0.84 (0.73–0.97)	0.018	
**Healthful Plant-based Diet Index (HPDI)**						
HPDI T1 (0–53)						0.001
Stair climbing (steps/day) None	301/2705	Reference		Reference		
Stair climbing (steps/day) 10–50	653/7156	1.03 (0.90–1.18)	0.668	1.01 (0.88–1.16)	0.843	
Stair climbing (steps/day) 60–100	1137/13,467	0.91 (0.80–1.03)	0.137	0.94 (0.83–1.07)	0.352	
Stair climbing (steps/day) 110–150	546/6889	0.86 (0.75–0.99)	0.033	0.91 (0.79–1.05)	0.197	
Stair climbing (steps/day) > 150	459/5323	0.96 (0.83–1.11)	0.617	1.04 (0.90–1.21)	0.571	
HPDI T2 (54–60)						
Stair climbing (steps/day) None	338/3025	1.02 (0.87–1.19)	0.836	1.06 (0.91–1.24)	0.475	
Stair climbing (steps/day) 10–50	635/7450	0.93 (0.81–1.07)	0.307	0.95 (0.83–1.10)	0.512	
Stair climbing (steps/day) 60–100	1242/15,551	0.86 (0.76–0.98)	0.020	0.92 (0.81–1.04)	0.200	
Stair climbing (steps/day) 110–150	645/8563	0.81 (0.71–0.93)	0.003	0.89 (0.77–1.02)	0.085	
Stair climbing (steps/day) > 150	506/6851	0.81 (0.70–0.93)	0.004	0.90 (0.78–1.04)	0.163	
HPDI T3 (61–62)						
Stair climbing (steps/day) None	277/3015	0.88 (0.75–1.04)	0.123	0.94 (0.80–1.11)	0.450	
Stair climbing (steps/day) 10–50	502/6671	0.86 (0.75–0.99)	0.042	0.91 (0.79–1.05)	0.190	
Stair climbing (steps/day) 60–100	1013/14,606	0.79 (0.70–0.90)	<0.001	0.86 (0.75–0.98)	0.021	
Stair climbing (steps/day) 110–150	634/8674	0.83 (0.72–0.95)	0.008	0.93 (0.81–1.06)	0.281	
Stair climbing (steps/day) > 150	520/7438	0.81 (0.70–0.94)	0.004	0.93 (0.80–1.07)	0.308	

Note: Model 1 was adjusted for age, sex, race/ethnicity, education and the Townsend deprivation index; model 2 was adjusted for model 1 plus drinking, smoking, total energy intake, physical activity, sleep duration, BMI, systolic blood pressure, diastolic blood pressure and baseline diabetes. Abbreviations: MACEs, major adverse cardiovascular events.

## Data Availability

The original contributions presented in the study are included in the article and Appendix A, further inquiries can be directed to the corresponding authors.

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
