# Peer review of "Association of Four Dietary Patterns and Stair Climbing with Major Adverse Cardiovascular Events: A Large Population-Based Prospective Cohort Study"

_nutrients, 2024, doi:10.3390/nu16213576_

Round 1
Reviewer 1 Report
Comments and Suggestions for Authors
This study, with a very large sample size of N=117,384 and a sufficiently long follow-up period, reports the association between MACE and stair climbing in a highly intriguing manner. The methodology is described with accuracy and sufficient detail. Additionally, Figure 1 clearly illustrates the non-linear relationship between the number of steps climbed and HR for MACE. The nutritional indices have been thoroughly examined, and the relationship between MACE and stair climbing has been meticulously analyzed across various patient groups with differing nutritional index levels.
Author Response
This study, with a very large sample size of N=117,384 and a sufficiently long follow-up period, reports the association between MACE and stair climbing in a highly intriguing manner. The methodology is described with accuracy and sufficient detail. Additionally, Figure 1 clearly illustrates the non-linear relationship between the number of steps climbed and HR for MACE. The nutritional indices have been thoroughly examined, and the relationship between MACE and stair climbing has been meticulously analyzed across various patient groups with differing nutritional index levels.
Response:
We thank the reviewer for the positive recognition of our work.
Reviewer 2 Report
Comments and Suggestions for Authors
-
The authors should provide a clear definition of MACE (Major Adverse Cardiovascular Events) in the methods section. As different studies define MACE in various ways, this can lead to increased heterogeneity and reduce the generalizability of the results. For reference, please consider the following articles:
https://pubmed.ncbi.nlm.nih.gov/29037211/
https://pubmed.ncbi.nlm.nih.gov/31992158/
https://pubmed.ncbi.nlm.nih.gov/28663251/ -
It is recommended that the authors conduct an additional regression model, adjusting for key confounders such as hypertension (HTN), hypercholesterolemia, diabetes mellitus (DM), obesity (BMI >30 kg/m²), smoking status (current or cessation within the past 3 months), positive family history of cardiovascular disease (parent or sibling with CVD before age 65), and atherosclerotic disease (previous MI, PCI/CABG, CVA/TIA, or peripheral arterial disease).
-
Could the authors elaborate on why dietary patterns alone do not appear to significantly reduce the risk of MACE, yet when stair climbing is added as a parameter, the association becomes significant?
-
The conclusion is not fully supported by the study’s findings. The authors state that adherence to healthy dietary patterns, particularly the AMED diet, combined with daily stair climbing, is linked to a reduced risk of MACE. However, Table 2 shows that adherence to the DASH diet combined with stair climbing reduces the risk of MACE more effectively than the Mediterranean diet (0.84, 95% CI: 0.73-0.97 vs. 0.86, 95% CI: 0.74-0.99, respectively). Please revise the conclusion to accurately reflect these results.
-
The authors should add a paragraph in the discussion section that explores the impact of high adherence to specific dietary patterns on the risk factors for MACE. Relevant studies for reference include:
https://pubmed.ncbi.nlm.nih.gov/25145972/
https://pubmed.ncbi.nlm.nih.gov/34352891/
https://pubmed.ncbi.nlm.nih.gov/37513679/
https://pubmed.ncbi.nlm.nih.gov/38816664/
Minor editing of English language required.
Author Response
Comment 1:
The authors should provide a clear definition of MACE (Major Adverse Cardiovascular Events) in the methods section. As different studies define MACE in various ways, this can lead to increased heterogeneity and reduce the generalizability of the results. For reference, please consider the following articles:
https://pubmed.ncbi.nlm.nih.gov/29037211/
https://pubmed.ncbi.nlm.nih.gov/31992158/
https://pubmed.ncbi.nlm.nih.gov/28663251/
Response 1:
We appreciate with this important comment. We have revised the manuscript including a clear definition of MACE.
Revised text in the manuscript:
Page 3, Materials and Methods: MACE was defined as incidence of nonfatal myocardial infarction, stroke, and fatal cardiovascular events.
Comment 2:
It is recommended that the authors conduct an additional regression model, adjusting for key confounders such as hypertension (HTN), hypercholesterolemia, diabetes mellitus (DM), obesity (BMI >30 kg/m²), smoking status (current or cessation within the past 3 months), positive family history of cardiovascular disease (parent or sibling with CVD before age 65), and atherosclerotic disease (previous MI, PCI/CABG, CVA/TIA, or peripheral arterial disease).
Response 2:
We sincerely thank the reviewer for this valuable suggestion. We have conducted additional analyses by adjusting for the aforementioned confounders. The results remain consistent with our main findings. For your convenience, we have copied the table below. Please refer to the Table S1 for further details.
|
Table S1. Association between dietary patterns combined with stair climbing and MACE. |
||||
|
  |
MACE |
|||
|
Dietary patterns and climbing stair |
Case/N |
HR (95% CI) |
P value |
P for interaction |
|
Alternate Mediterranean Diet score (AMED) |
|
|
|
<0.001 |
|
AMED T1 (0-3) |
|
|
|
|
|
Stair climbing (steps/day) None |
271/2389 |
Reference |
|
|
|
Stair climbing (steps/day) 10-50 |
577/6312 |
0.97 (0.84 - 1.12) |
0.702 |
|
|
Stair climbing (steps/day) 60-100 |
970/11523 |
0.91 (0.79 - 1.04) |
0.160 |
|
|
Stair climbing(steps/day) 110-150 |
463/6012 |
0.86 (0.74 - 1) |
0.050 |
|
|
Stair climbing(steps/day) >150 |
358/4725 |
0.87 (0.74 - 1.02) |
0.086 |
|
|
AMED T2 (4-5) |
|
|
|
|
|
Stair climbing (steps/day) None |
276/2598 |
0.9 (0.76 - 1.07) |
0.229 |
|
|
Stair climbing (steps/day) 10-50 |
545/6452 |
0.85 (0.74 - 0.99) |
0.034 |
|
|
Stair climbing (steps/day) 60-100 |
1052/13064 |
0.84 (0.74 - 0.97) |
0.014 |
|
|
Stair climbing(steps/day) 110-150 |
573/7168 |
0.86 (0.74 - 1) |
0.044 |
|
|
Stair climbing(steps/day) >150 |
443/5625 |
0.9 (0.77 - 1.05) |
0.173 |
|
|
AMED T3 (6-9) |
|
|
|
|
|
Stair climbing (steps/day) None |
369/3758 |
0.87 (0.74 - 1.01) |
0.073 |
|
|
Stair climbing (steps/day) 10-50 |
668/8513 |
0.84 (0.73 - 0.96) |
0.014 |
|
|
Stair climbing (steps/day) 60-100 |
1370/19037 |
0.77 (0.68 - 0.88) |
<0.001 |
|
|
Stair climbing(steps/day) 110-150 |
789/10946 |
0.79 (0.69 - 0.91) |
0.001 |
|
|
Stair climbing(steps/day) >150 |
684/9262 |
0.85 (0.73 - 0.98) |
0.023 |
|
|
Alternate Healthy Eating Index-2010 (AHEI-2010) |
|
|
|
<0.001 |
|
AHEI-2010 T1 (0-47) |
|
|
|
|
|
Stair climbing (steps/day) None |
340/2847 |
Reference |
|
|
|
Stair climbing (steps/day) 10-50 |
700/7537 |
0.94 (0.82 - 1.07) |
0.340 |
|
|
Stair climbing (steps/day) 60-100 |
1251/14597 |
0.88 (0.78 - 0.99) |
0.037 |
|
|
Stair climbing(steps/day) 110-150 |
585/7574 |
0.81 (0.71 - 0.93) |
0.002 |
|
|
Stair climbing(steps/day) >150 |
476/5866 |
0.89 (0.78 - 1.03) |
0.115 |
|
|
AHEI-2010 T2 (48-58) |
|
|
|
|
|
Stair climbing (steps/day) None |
292/2833 |
0.89 (0.76 - 1.05) |
0.163 |
|
|
Stair climbing (steps/day) 10-50 |
551/6863 |
0.86 (0.75 - 0.98) |
0.027 |
|
|
Stair climbing (steps/day) 60-100 |
1089/14275 |
0.83 (0.74 - 0.94) |
0.003 |
|
|
Stair climbing(steps/day) 110-150 |
645/7945 |
0.91 (0.8 - 1.04) |
0.170 |
|
|
Stair climbing(steps/day) >150 |
480/6318 |
0.88 (0.77 - 1.01) |
0.074 |
|
|
AHEI-2010 T3 (59-95) |
|
|
|
|
|
Stair climbing (steps/day) None |
284/3065 |
0.9 (0.77 - 1.06) |
0.205 |
|
|
Stair climbing (steps/day) 10-50 |
539/6877 |
0.9 (0.78 - 1.03) |
0.129 |
|
|
Stair climbing (steps/day) 60-100 |
1052/14752 |
0.83 (0.74 - 0.94) |
0.004 |
|
|
Stair climbing(steps/day) 110-150 |
595/8607 |
0.83 (0.73 - 0.95) |
0.007 |
|
|
Stair climbing(steps/day) >150 |
529/7428 |
0.9 (0.78 - 1.03) |
0.121 |
|
|
Dietary Approaches to Stop Hypertension (DASH) |
|
|
|
<0.001 |
|
DASH T1 (0-20) |
|
|
|
|
|
Stair climbing (steps/day) None |
320/2743 |
Reference |
|
|
|
Stair climbing (steps/day) 10-50 |
661/7429 |
0.94 (0.82 - 1.07) |
0.347 |
|
|
Stair climbing (steps/day) 60-100 |
1142/13976 |
0.87 (0.77 - 0.98) |
0.027 |
|
|
Stair climbing(steps/day) 110-150 |
581/7239 |
0.89 (0.78 - 1.02) |
0.099 |
|
|
Stair climbing(steps/day) >150 |
444/5501 |
0.93 (0.81 - 1.08) |
0.329 |
|
|
DASH T2 (21-25) |
|
|
|
|
|
Stair climbing (steps/day) None |
284/2834 |
0.86 (0.74 - 1.02) |
0.076 |
|
|
Stair climbing (steps/day) 10-50 |
620/7025 |
0.91 (0.8 - 1.05) |
0.189 |
|
|
Stair climbing (steps/day) 60-100 |
1192/14488 |
0.87 (0.77 - 0.99) |
0.032 |
|
|
Stair climbing(steps/day) 110-150 |
609/8050 |
0.82 (0.72 - 0.94) |
0.005 |
|
|
Stair climbing(steps/day) >150 |
507/6424 |
0.9 (0.78 - 1.04) |
0.145 |
|
|
DASH T3 (26-40) |
|
|
|
|
|
Stair climbing (steps/day) None |
312/3168 |
0.92 (0.79 - 1.08) |
0.311 |
|
|
Stair climbing (steps/day) 10-50 |
509/6823 |
0.83 (0.72 - 0.96) |
0.010 |
|
|
Stair climbing (steps/day) 60-100 |
1058/15160 |
0.79 (0.7 - 0.9) |
0.000 |
|
|
Stair climbing(steps/day) 110-150 |
635/8837 |
0.83 (0.72 - 0.95) |
0.007 |
|
|
Stair climbing(steps/day) >150 |
534/7687 |
0.84 (0.73 - 0.96) |
0.013 |
|
|
Healthful Plant-based Diet Index (HPDI) |
|
|
|
<0.001 |
|
HPDI T1 (0-53) |
|
|
|
|
|
Stair climbing (steps/day) None |
301/2705 |
Reference |
|
|
|
Stair climbing (steps/day) 10-50 |
653/7156 |
1.01 (0.88 - 1.16) |
0.867 |
|
|
Stair climbing (steps/day) 60-100 |
1137/13467 |
0.94 (0.83 - 1.07) |
0.336 |
|
|
Stair climbing(steps/day) 110-150 |
546/6889 |
0.91 (0.79 - 1.05) |
0.209 |
|
|
Stair climbing(steps/day) >150 |
459/5323 |
1.04 (0.9 - 1.2) |
0.598 |
|
|
HPDI T2 (54-60) |
|
|
|
|
|
Stair climbing (steps/day) None |
338/3025 |
1.05 (0.9 - 1.23) |
0.538 |
|
|
Stair climbing (steps/day) 10-50 |
635/7450 |
0.95 (0.83 - 1.09) |
0.500 |
|
|
Stair climbing (steps/day) 60-100 |
1242/15551 |
0.92 (0.81 - 1.05) |
0.215 |
|
|
Stair climbing(steps/day) 110-150 |
645/8563 |
0.89 (0.77 - 1.02) |
0.091 |
|
|
Stair climbing(steps/day) >150 |
506/6851 |
0.9 (0.78 - 1.04) |
0.159 |
|
|
HPDI T3 (61-62) |
|
|
|
|
|
Stair climbing (steps/day) None |
277/3015 |
0.94 (0.8 - 1.11) |
0.453 |
|
|
Stair climbing (steps/day) 10-50 |
502/6671 |
0.92 (0.8 - 1.06) |
0.250 |
|
|
Stair climbing (steps/day) 60-100 |
1013/14606 |
0.86 (0.75 - 0.98) |
0.023 |
|
|
Stair climbing(steps/day) 110-150 |
634/8674 |
0.93 (0.81 - 1.07) |
0.297 |
|
|
Stair climbing(steps/day) >150 |
520/7438 |
0.93 (0.81 - 1.08) |
0.337 |
  |
|
Note: Model was adjusted for age, sex, race/ethnicity, education, Townsend deprivation index, drinking status, smoking status, total energy intake, physical activity, sleep duration, BMI, systolic blood pressure, diastolic blood pressure, diabetes, hypertension, hypercholesterolemia, obesity (BMI >30kg/m2), atherosclerotic disease, and history of cardiovascular disease. Abbreviations: MACE, major adverse cardiovascular events. |
||||
Comment 3:
Could the authors elaborate on why dietary patterns alone do not appear to significantly reduce the risk of MACE, yet when stair climbing is added as a parameter, the association becomes significant?
Response 3:
Thank you for the insightful question. Both diet and stair climbing may have a synergistic effect on cardiovascular health. Combining a healthy diet with regular physical activity leads to more comprehensive improvements in health outcomes, significantly reducing risk of major adverse cardiovascular events (MACE). Physical activity enhances the beneficial effects of a healthy diet by improving metabolism, insulin sensitivity, and lipid profiles. Individuals who engage in regular stair climbing tend to be more physically active overall, which may further lower the risk of MACE. This additional level of physical activity could be the key factor that makes the association significant, as diet alone may not adequately capture the full benefits of a healthy lifestyle.
Comment 4:
The conclusion is not fully supported by the study’s findings. The authors state that adherence to healthy dietary patterns, particularly the AMED diet, combined with daily stair climbing, is linked to a reduced risk of MACE. However, Table 2 shows that adherence to the DASH diet combined with stair climbing reduces the risk of MACE more effectively than the Mediterranean diet (0.84, 95% CI: 0.73-0.97 vs. 0.86, 95% CI: 0.74-0.99, respectively). Please revise the conclusion to accurately reflect these results.
Response 4:
Thank you for your suggestion. In the total population, we observed that adherence to the AMED and DASH, combined with stair climbing, reduced the risk of MACE to varying degrees. However, the reference populations for these two risk ratios are not the same, which limits our ability to directly compare the differences in dietary patterns using hazard ratios (HR). Our conclusions are based on results from both the general population and obese participants, indicating that AMED combined with stair climbing is associated with a lower risk of MACE compared to AMED without stair climbing. Nonetheless, your suggestion is valuable, and we acknowledge that we lack the most direct measure to fully capture the advantages of the AMED. In response to your feedback, we have revised our conclusions accordingly.
Comment 5:
The authors should add a paragraph in the discussion section that explores the impact of high adherence to specific dietary patterns on the risk factors for MACE. Relevant studies for reference include:
https://pubmed.ncbi.nlm.nih.gov/25145972/
https://pubmed.ncbi.nlm.nih.gov/34352891/
https://pubmed.ncbi.nlm.nih.gov/37513679/
https://pubmed.ncbi.nlm.nih.gov/38816664/
Response 5:
Author response: We thank the reviewer for highlighting this important point. We have added a paragraph in the discussion section accordingly.
Revised in the manuscript:
Page 17, Discussion: High adherence to healthy dietary patterns such as AMED or DASH reduces systemic inflammation levels, improves metabolic disorders (obesity, diabetes) and thus reduces the occurrence of more severe MACE. These healthy diets promote the intake of essential micronutrients and bioactives that not only improve cardiovascular and cerebrovascular metabolic health in addition to reducing cellular oxidative stress, but also help to slow down the aging process. These findings emphasize the public health importance of a high degree of adherence to healthy dietary patterns reducing the huge economic burden of disease and greatly improving quality of life.
Reviewer 3 Report
Comments and Suggestions for Authors
The study addresses an important population problem.
The study is well-planned, well-conducted, well-described, and well-discussed.
The conclusions are reasonable.
I have only minor suggestions:
1. The introduction is written very generally. More epidemiological data illustrating the statements should be added.
2. A separate subsection on the study objectives should be added.
3. Justification for the choice of methods for assessing dietary habits should be added.
4. Data from the manufacturer of the statistical application R should be added.
5. Figures 3 and 4 should be corrected, they are illegible.
Author Response
Comment 1:
The introduction is written very generally. More epidemiological data illustrating the statements should be added.
Response 1:
We agree with the reviewer’s suggestion. The introduction has been rewritten to include more epidemiological data (Page 2).
Comment 2:
A separate subsection on the study objectives should be added.
Response 2:
Thank you for your comment. We have added a separate subsection on the study objectives in the revised manuscript.
Revised in the manuscript:
Page 2, Introduction: To date, only a few studies have investigated the interactive or combined associations between dietary patterns and stair climbing, with limited subset having explored CVD outcomes. To investigate the cardiovascular protective effects of these two healthy lifestyles. In this study, we investigated the individual and combined effects of four healthy dietary patterns (AMED, AHEI, HPDI, DASH), daily stair climbing frequency on MACE in a large prospective cohort study.
Comment 3:
Justification for the choice of methods for assessing dietary habits should be added.
Response 3:
Thank you for pointing out this. We have illustrated that diet was assessed using a web-based 24-hour dietary assessment tool (the vali-dated Oxford WebQ). A sub-cohort of the UK Biobank completed the assessment on ≥1 of the five occasions between April 2009 and June 2012. The Oxford WebQ has been validated against an interviewer administered 24-hour dietary recall, producing a mean Spearman correlation coefficient for macronutrients of 0.62 (range 0.54-0.69). The methods chosen for assessing dietary habits were based on their validated use in previous large-scale epidemiological studies, ensuring both reliability and reproducibility. In this study, in order to estimate usual intake, the average food/nutrient intakes of the two or more dietary assessments were used in the analysis.
Comment 4:
Data from the manufacturer of the statistical application R should be added.
Response 4:
Thank you for your comment. We have included the relevant data from the manufacturer of the statistical application R as suggested.
Revised in the manuscript:
Page 5, Materials and Methods: Statistical analyses were conducted with R, version 4.2.1 (R Foundation for Statistical Computing, Vienna, Austria). A 2-sided P < 0.05 was considered statistically significant.
Comment 5:
Figures 3 and 4 should be corrected, they are illegible.
Response 5:
Thank you for your comment. We have made the corrections as suggested. For your convenience, we have copied the figures below; please refer to Figure 3 and Figure 4 for details.
Round 2
Reviewer 2 Report
Comments and Suggestions for Authors
The authors have addressed my remarks in a thorough and satisfactory fashion.